# Relationships among Physical Activity, Pain, and Bone Health in Youth and Adults with Thalassemia: An Observational Study

Ellen B. Fung *, Elijah K. Goldberg, Sakina Bambot, Raquel Manzo and Ashutosh Lal

Division of Hematology, Department of Pediatrics, UCSF Benioff Children's Hospital Oakland, Oakland, CA 94609, USA
* Correspondence: ellen.fung@ucsf.edu

**Abstract:** Patients with thalassemia (Thal) engage in less physical activity than non-Thal populations, which may contribute to pain and osteoporosis. The purpose of this study was to assess relationships between physical activity, pain, and low bone mass in a contemporary sample of patients with Thal. Seventy-one patients with Thal (50 adults $\geq$18 years, 61% male, 82% transfusion-dependent) completed the Brief Pain Inventory Short Form and validated physical activity questionnaires for youth and adults. Nearly half of the patients reported daily somatic pain. Using multiple regression, after controlling for age and gender, sedentary behavior was positively associated with pain severity ($p = 0.017$, $r^2 = 0.28$). Only 37% of adult participants met CDC recommendations for physical activity. Spine BMD Z-score was higher ($-2.1 \pm 0.7$) in those who met activity guidelines compared to those who did not ($-2.8 \pm 1.2$, $p = 0.048$). A positive relationship was observed between self-reported physical activity (hours/week) and hip BMD Z-score in adults with Thal after controlling for transfusion status and sedentary activity time ($p = 0.009$, $r^2 = 0.25$). These results suggest that decreased physical activity and increased sedentary behavior contribute to low bone mass, which may be related to pain severity in some patients with Thal. Studies focused on increasing physical activity may contribute to improved bone health and reduced pain in patients with Thal.

**Keywords:** physical activity; thalassemia; sedentary behavior; BMD; pain; low bone mass



## 1. Introduction

Patients with thalassemia (Thal) suffer from a significant number of comorbid conditions including fatigue [1], cardiomyopathy [2], body pain [3,4], diabetes [5], and low bone mass or osteoporosis [6]. A mainstay of therapy for many patients with Thal is regular red blood cell transfusion to reduce the disease characteristics associated with severe anemia [7]. However, without adequate chelation, transfusion therapy frequently results in excessive organ iron deposition, leading to systemic toxicity. Over the past few decades, life expectancy for patients with Thal has increased due to the optimization of red blood cell transfusions and iron chelation treatments. Following this decrease in early iron-related mortality, clinicians are exploring ways to improve quality of life and reduce pain for patients with Thal.

Previous research conducted in small cohorts has suggested that patients with Thal have reduced physical activity compared to individuals without Thal [8–10]. Anemia, fatigue, and deconditioning, all of which are frequently observed in patients with Thal, can hamper an individual from exercising. It is hypothesized that significant cardiac iron overload can further decrease exercise capacity by reducing stroke volume, cardiac contractility, and preventing exercise-induced increases in heart rate [9,11,12]. Chronic pain, which is reported in over half of adult patients with Thal, has also been shown to affect physical function and encourage sedentary behaviors, such as prolonged sitting and watching television [4]. These resultant sedentary behaviors may contribute to osteoporosis, diabetes, depression, and a decreased quality of life.

The potential benefits of physical activity for individuals with Thal cannot be ignored. In non-Thal populations, increased bone strength [13], enhanced muscle and bone growth [14,15], improved glucose tolerance [16], reduced pain [17], improvements in mood [18], and academic performance [19] have been consistently reported in relation to increased physical activity. The assessment of exercise habits in patients with Thal is critical to further our understanding of how daily sedentary behavior and patterns of inactivity may be associated with frequently reported comorbid conditions.

The purpose of this study was to assess self-reported physical activity in a contemporary sample of youth and adults with thalassemia and to explore associations with levels of pain, bone mass, and antidepressant medication usage. This is the first study to gather quantitative data on exercise patterns in patients with thalassemia and explore relationships between these habits and frequently exhibited comorbidities in this at-risk population.

## 2. Materials and Methods

This was an observational, cross-sectional, convenience study of self-reported physical activity and pain in patients with thalassemia >5 years of age. Study objectives were explained to subjects during a regularly scheduled visit to the Thalassemia Comprehensive Clinic at the UCSF Benioff Children's Hospital Oakland. If interested in participating, patients and/or caregivers reviewed and signed the written consent form, while children and adolescents (5 to 17 years) signed an assent form. The study was reviewed and approved by the Institutional Review Board at our center.

For the purposes of this report, youth were subjects <18 years of age. Each participating subject, regardless of age, first completed the validated Brief Pain Inventory Short Form (BPI-SF) [20–23]. Youth were then asked to complete the Family Nutrition and Physical Activity Screening Tool (FNPA) [24,25]. Instead of the FNPA, adults (≥18 years) completed the Sedentary Behavior Questionnaire (SBQ) [26] and the Physical Activity Assessment Tool [27]. The questionnaires were scored as described below.

The BPI-SF asked subjects a series of questions related to the severity, location, and impact of any pain that they may experience. Acute pain prevalence, location of pain, pain medication use, and medication-related pain relief were determined. Four questions asked subjects to rank their pain at its most and least severe in the last 24 h, their average pain, and their current pain on a scale from 0 (no pain) through 10 (pain as bad as you can imagine). These were added to obtain a pain severity score (range 0 to 40). Pain severity was further categorized into mild (pain severity score: 0 to 12), moderate (13 to 26), or severe pain (27 to 40). Additional questions were asked about the extent to which subjects' pain interfered with general activity, mood, walking ability, normal work, relations with other people, sleep, and enjoyment of life. These points were added to obtain a second pain interference score (range 0 to 70). Subjects were also given a body diagram and asked to mark locations where they had felt pain that day and then asked to list what medications or treatments they were currently taking for their pain.

The FNPA Screening Tool asked caregivers of youth subjects to answer a series of questions related to their child and family's dietary habits, screen time, physical activity, and sleeping behaviors. For youth >12 years of age, parents and youth cooperated to complete the survey. A Likert scale was used for each question: never/almost never (1), sometimes (2), often (3), or very often/always (4). For fourteen questions, a higher number corresponded to healthier behaviors, while for six questions, a higher number corresponded to less healthy behaviors. Therefore, these six questions were reverse-scored when tabulating a total FNPA score.

For adult subjects, the SBQ asked subjects to estimate how much time they spent in various sedentary behaviors, including playing video games, engaging in screen time, reading, doing inactive hobbies, and engaging in other seated tasks, on weekdays and weekends, ranging from 0 to >6 h. The total number of minutes was added to obtain the average amount of time subjects spent in sedentary behaviors during weekdays, weekends, and the entire week.

Lastly, the Physical Activity Assessment Tool quantified self-reported physical activity behaviors in adult subjects. The average number of days and minutes per day subjects spent in moderate levels of physical activity were reported for the previous 7 days. The number of days was multiplied by the number of minutes per day to obtain the number of minutes of moderate exercise per week. The same scoring was performed for time spent in vigorous level of physical activity. Subjects were also asked if the level of activity they reported for the last 7 days was more, less, or about the same as the last 3 months. Additional questions focused on motivations around exercise and barriers to participation in exercise. Physical characteristics, thalassemia phenotype, chronic transfusion, use of antidepressant medication or cigarettes, 25OH vitamin D level, and bone mineral density (BMD) Z-scores were abstracted from the most recent information contained in the patient's medical record. Circulating 25OH vitamin D of <30 ng/mL was defined as insufficient. BMD Z-scores at any skeletal site $\leq -2.0$ were defined as low bone mass [28,29]. The adult and youth data were analyzed separately due to the use of disparate validated questionnaires. For the adult cohort, activity levels, summarized as minutes per week, were compared to the Center for Disease Control (CDC) recommendations for physical activity for adults [30].

*Statistics*

Summary analyses of all variables were first reviewed for outliers using scatterplots, means, and standard deviations. Continuous variables were explored using *t*-tests and categorical variables were analyzed using chi-squared tests, with Fischer's exact follow-up for small samples. Multivariate linear and or logistic regression statistics were used to explore predictors of pain and bone health. STATA (StatCorp, College Station, TX, USA, v. 16.0) was used for all statistical analyses.

## 3. Results

A total of 71 subjects with Thal participated in this observational study, including 43 males and 50 adults. Characteristics and demographics are reported in Table 1. As expected, body mass index (BMI) was higher for adults compared to youth ($p = 0.001$); only 10% of patients were considered underweight (BMI <18.5 kg/m$^2$), while 20% were considered overweight or obese (BMI > 25.0 kg/m$^2$). Vitamin D insufficiency was more common in youth subjects compared to adult subjects ($p = 0.001$), but not different by transfusion status.

**Table 1.** Demographics and disease characteristics in youth ($\geq$5 and <18 years) and adult ($\geq$18 years) patients with thalassemia.

| | Youth<br>$n = 21$ | Adults<br>$n = 50$ | Total<br>$n = 71$ | *p*-Value [2] |
|---|---|---|---|---|
| Age, years | 11.5 $\pm$ 3.6 [1]<br>(5.7–17.5) | 34.1 $\pm$ 9.4<br>(18.7–58.3) | 27.4 $\pm$ 13.2 | <0.001 |
| Male/female | 13/8 | 30/20 | 43/28 | NS |
| Thalassemia type, n | β-Thal, 12<br>E-ß Thal, 3<br>HbH or H/CS, 5<br>Other, 1 | β-Thal, 24<br>E-β Thal, 15<br>HbH or H/CS, 8<br>Other, 3 | β-Thal, 36<br>E-β Thal, 18<br>HbH or H/CS, 13<br>Other, 4 | NS |
| TDT, NTDT [3] | 71.4%, 28.6% | 86.0%, 14.0% | 81.7%, 18.3% | 0.15 |
| Body mass index, kg/m$^2$ | 17.6 $\pm$ 3.1 | 22.2 $\pm$ 3.3 | 20.8 $\pm$ 3.8 | <0.001 |
| Overweight or obese [4] | 4.7% | 10% | 8.5% | NS |
| Current smoker [5] | 0 | 8.0% | 5.6% | NS |
| Former smoker | 0 | 6.0% | 4.2% | NS |
| Prescribed antidepressant medication | 0 | 12% | 6.5% | 0.11 |

**Table 1.** *Cont.*

|  | Youth $n = 21$ | Adults $n = 50$ | Total $n = 71$ | *p*-Value [2] |
|---|---|---|---|---|
| 25OH Vit D, ng/mL | $28.2 \pm 10.9$ | $38.8 \pm 15.3$ | $35.6 \pm 14.9$ | 0.008 |
| Low vitamin D [6] | 52.6% | 24.4% | 32.8% | 0.028 |
| TDT | 50% | 63.6% | 57.4% | NS [7] |
| NTDT | 50% | 36.4% | 42.8% | |
| Spine BMD Z-score | $-1.1 \pm 0.7$ | $-2.5 \pm 1.2$ | $-2.1 \pm 1.2$ | <0.001 |
| Hip BMD Z-score [8] | . | $-1.7 \pm 1.1$ | . | |
| Whole body less head [8] BMD Z-score | $-1.8 \pm 0.5$ | . | . | |
| Low bone mass [9] | 41.2% | 69.4% | 62.1% | 0.039 |
| TDT | 85.7% | 88.2% | 87.8% | NS [10] |
| NTDT | 14.3% | 11.8% | 12.2% | |

[1] Continuous variables are presented as the mean ± standard deviation (SD). Categorical variables are included as the prevalence (%). [2] *p*-Values are reported for either *t*-test (continuous variables) or chi-square test (categorical variables) between youth and adult cohorts. [3] TDT: transfusion-dependent; NTDT: non-transfusion-dependent. [4] Overweight or obese in adults is defined as the prevalence of individuals with a body mass index >25.0 mg/kg$^2$ in youth BMI >95%. [5] Fischer's exact test was used for chi-squared analyses with small sample size. [6] Low vitamin D was defined as <30 ng/mL. [7] *p*-Values for prevalence of age by transfusion status were restricted to those subjects with low vitamin D level (chi-squared statistic). [8] Hip DXA scans were not performed in youth, while whole-body DXA scans were not performed in adult patients. [9] Low bone mass was defined as any BMD Z-score $\le -2.0$. [10] *p*-Values for prevalence of age by transfusion status were restricted to those subjects with low bone mass (chi-squared statistic).

### 3.1. Bone Health

As has been observed previously, more adults with Thal had low bone mass (BMD Z-score $\le -2.0$) compared to youth (69% vs. 41%, $p = 0.039$). Low bone mass prevalence did not differ by gender, smoking status, vitamin D sufficiency, or type of thalassemia. Transfusion-dependent patients (TDT) tended to have a higher prevalence of low bone mass compared to non-transfusion-dependent patients (NTDT) (67% vs. 42%, $p = 0.1$). All of the patients ($n = 6$) who were on antidepressant medications also had low bone mass. In youth with Thal, when parents answered the question, '*How often does my child do something physically active when they have free time?*', none of the subjects who answered 'always' had low bone mass, while 75% of those who reported 'never' had low bone mass.

Using multiple regression analyses for the whole cohort, spine BMD Z-score was negatively associated with age ($p < 0.001$), but positively associated with pain severity ($p = 0.023$, $r^2 = 0.21$). There was a trend toward a relationship between physical activity (hours/week) and spine BMD Z-score after controlling for transfusion status and sedentary behavior ($p = 0.055$, $r^2 = 0.15$). A significant positive relationship was observed between physical activity in adults (hours/week) and hip BMD Z-score after controlling for transfusion status and sedentary behavior ($p = 0.009$, $r^2 = 0.25$).

### 3.2. Pain

Forty-seven percent of all subjects experienced pain in the 24 h prior to completing the BPI-SF (Table 2). No difference in pain prevalence was observed by gender, thalassemia type, transfusion status, or vitamin D sufficiency. There was a trend toward more adults experiencing pain (54%) compared to youth (29%, $p = 0.059$). There was also a trend toward those with daily pain being on antidepressants ($p = 0.059$). Pain severity was strongly associated with interference in general activities, normal work, walking ability, mood, and enjoyment of life (all $p < 0.001$). Although the presence of pain was similar between males and females, females reported more severe pain (pain severity score: 10.1 females vs. 5.7 males, $p = 0.024$) and were taking prescription medications for pain (18% females vs. 5% males, $p = 0.01$). Reports of pain interfering with daily activities was associated with increasing BMI ($p = 0.007$) after controlling for age. In youth with Thal, increased healthy behaviors (as defined by a lower total FNPA score) were related to less severe pain ($p = 0.001$).

**Table 2.** Self-reported pain prevalence and pain characteristics in youth (≥5 and <18 years) and adult (≥18 years) patients with thalassemia.

| | Youth *n* = 21 | Adults *n* = 50 | Total *n* = 71 | *p*-Value [1] |
|---|---|---|---|---|
| Pain in past 24 h, count (% of cohort) | 6 (28.6) | 27 (54.0) | 33 (46.5) | 0.050 |
| Back pain, count (%) | 5 (23.8) | 22 (44.9) | 27 (38.6) | 0.097 |
| Pain in other body areas, count (%) | 6 (28.6) | 19 (38.8) | 25 (35.7) | NS |
| Any pain medication use, count (%) | 5 (23.8) | 23 (46.0) | 28 (39.4) | 0.08 |
| Average level of pain, 1–10 scale | 1.8 ± 2.3 | 2.6 ± 2.5 | 2.3 ± 2.4 | NS |
| Pain severity score [2] | 6.3 ± 7.8 | 7.8 ± 8.4 | 7.4 ± 8.2 | NS |
| Pain severity category [3] | Mild: 33% Moderate: 67% Severe: 0% | Mild: 59% Moderate: 30% Severe: 11% | Mild: 55% Moderate: 36% Severe: 9% | NS |
| Pain interference score [4] | 13.4 ± 16.0 | 13.6 ± 17.8 | 13.6 ± 17.2 | NS |

[1] *p*-Values are reported for differences between youth and adult cohorts. [2] Pain severity score: summary of four pain severity questions, on a scale of 0 (min) to 40 (max). [3] Pain severity categories in those who experienced pain in previous 24 h (*n* = 33) are defined by pain severity score as mild (pain severity score: 0–12), moderate (13–26), or severe (27–40). [4] Pain interference score: summary of seven questions related to how pain interferes with daily activities and mood, on a scale of 0 (min) to 70 (max).

### 3.3. Activity

In adults, the three most commonly reported reasons for exercising included for health, to feel better, and for weight control. 15% of adults reported no moderate or vigorous physical activity in the past week, 21% of subjects reported engaging in moderate or vigorous activity 3 to 5 days per week, and 4% reported daily moderate or vigorous physical activity (Table 3). BMI was not different between those who reported exercising daily in the past week compared to those who did not exercise at all. Only 37% of adults reported participating in moderate levels of physical activity for a minimum of 30 min per day for 5 days out of the week, the CDC recommendation for physical activity. Spine BMD Z-score was higher (−2.1 ± 0.7) in those who met activity guidelines compared to those who did not (−2.8 ± 1.2, *p* = 0.048, Figure 1). Whether or not a subject met the activity guideline was not related to their gender, transfusion status, thalassemia type, or pain prevalence. Using multiple regression and controlling for age and gender, sedentary behavior was positively associated with pain severity (*p* = 0.017, $r^2$ = 0.28).

**Table 3.** Self-reported physical activity and sedentary behavior in adult subjects with thalassemia (*n* = 51, ≥18 years).

| | Mean ± SD | Range |
|---|---|---|
| Time spent in moderate physical activity for at least 10 min (days per week) | 3.2 ± 2.2 | 0–7 |
| Time spent in moderate physical activity (minutes per week) | 178 ± 211 | 0–870 |
| Time spent in vigorous physical activity (minutes per week) | 57 ± 91 | 0–300 |

**Table 3.** *Cont.*

| | Mean ± SD | Range |
|---|---|---|
| *Average time spent in physical activity (moderate + vigorous, hours per day)* | 0.5 ± 0.7 | 0–2.7 |
| Time spent in sedentary behaviors on weekdays (minutes) | 574 ± 266 | 80–1170 |
| Time spent in sedentary behaviors on weekends (minutes) | 521 ± 278 | 60–1260 |
| *Average time spent in sedentary behaviors (hours per day)* | 2.6 ± 1.2 | 0.6–5.4 |

Responses to the Physical Activity Assessment Tool [27] which reports on physical activity in previous 7 days. Responses to the Sedentary Behavior Questionnaire [26] which reports upon behavior in previous 30 days. CDC physical activity guideline: Minimum of 30 min of moderate-level physical activity 5 days per week or a total of 150 min per week [30] (Physical Activity Guidelines for Americans, second edition).

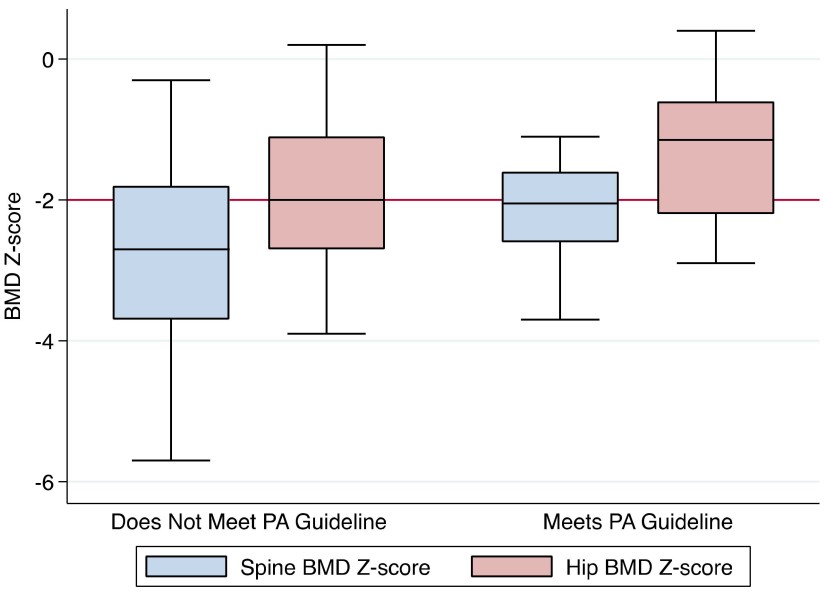

**Figure 1.** Spine and hip BMD Z-scores in adult subjects with thalassemia, stratified by success or failure to meet CDC physical activity guidelines (*n* = 51). Spine BMD Z-score comparison between those who met the CDC guideline for physical activity per week (30 min/day, 5 days per week or 150 min, −2.1 ± 0.7) and those who did not (−2.8 ± 1.3; *p* = 0.048). Hip BMD Z-score comparison between those who met the PA guideline (−1.4 ± 0.9) and those who did not (−1.9 ± 1.2; *p* = 0.096). Low bone mass defined by reference line placement at BMD Z-score = −2.0.

*3.4. Healthy Behaviors in Youth*

The FNPA was used to capture parent reports of food and beverage choices, family meals, screen time, physical activity, and sleep routine in youth with Thal. The questions which pertained to behaviors related to physical activity are summarized in Figure 2. 81% of youth reported often having more than 2 h of screen time per day, while only 29% frequently participated in organized sports or physical activities. The mean FNPA score, reflecting the summary of healthy behaviors, was 23.2 ± 12.1 on a scale from 0 to 70. When a subset of Thal patients were compared to previously published data from a cohort of healthy children [31] (age range: 6 to 10 years), the FNPA score was significantly lower in youth with Thal (*n* = 8, 30.1 ± 10.6) than in healthy children (*n* = 714; 65.9 ± 6.1; *p* < 0.001).

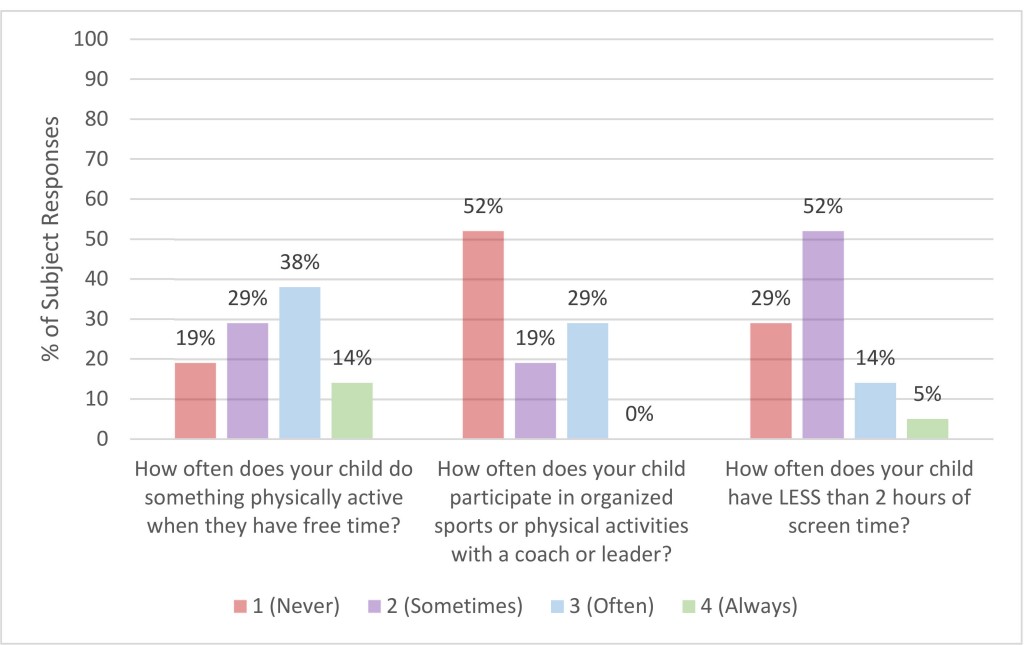

**Figure 2.** Parent-reported physical activity behaviors in youth (≥5 and <18 years) with thalassemia (*n* = 21). Average responses for youth with thalassemia from the Family Nutrition and Physical Activity Screening Tool (FNPA).

## 4. Discussion

In this contemporary sample of youth and adults with thalassemia, more than half of adults with Thal experienced daily pain and over two-thirds had low bone mass. Similar to previous reports, both pain and low bone mass increased with the age of patients [6,32], and sedentary behavior was commonly reported in both youth and adults [8]. Unique findings included the observed relationships between chronic pain, self-reported physical activity, and bone health (Figure 3). In this study, pain severity was related to more time spent in sedentary behavior. Sedentary behavior is typically inversely related to time spent exercising. In this cohort, physical activity was positively associated with BMD at a weightbearing skeletal site, as seen in hip BMD Z-score, those who followed activity guidelines had higher spine BMD Z-scores.

Prior studies which assessed physical activity in cohorts with Thal did not include measures of bone health [8–11]. In the most comprehensive study of bone health in patients with Thal to date, the results from a single self-reported question regarding physical activity level was not associated with spine or hip BMD Z-score, although a weak correlation with whole body bone mineral content was observed [6]. A more quantitative assessment of physical activity was used in this study. Although we believe this to be the first report of a relationship between bone health and activity in Thal, the finding is not surprising. Weightbearing physical activity is repeatedly shown to enhance bone mass, particularly in prepubertal populations [33], and exercise is known to protect against bone loss in aging non-Thal populations [34].

Similar to previous reports regarding pain prevalence, patients in this cohort reported frequent chronic pain which interfered with their daily activities [4,35]. The etiology of pain in thalassemia is unknown but is clearly multifactorial. Previous reports were unable to link pain to low hemoglobin, iron overload, or low bone mass [4]. Pain prevalence has been associated with a decreased quality of life, as well as less physical activity, in a large cohort of US patients with Thal [4]. In this study, pain severity was associated with a greater spine BMD Z-score after controlling for age. Similarly, in the 2013 study by Haines, individuals reporting daily pain were found to have significantly greater spine BMD Z-scores (*p* = 0.009) compared to patients with no pain, which was unrelated to

body weight [3]. It remains unclear as to why pain may affect skeletal sites differently. Abnormal vertebral morphometry is commonly observed in Thal [6] and is related to back pain. Subclinical vertebral compression fractures could explain the observed relationship between pain severity and increased spine BMD.

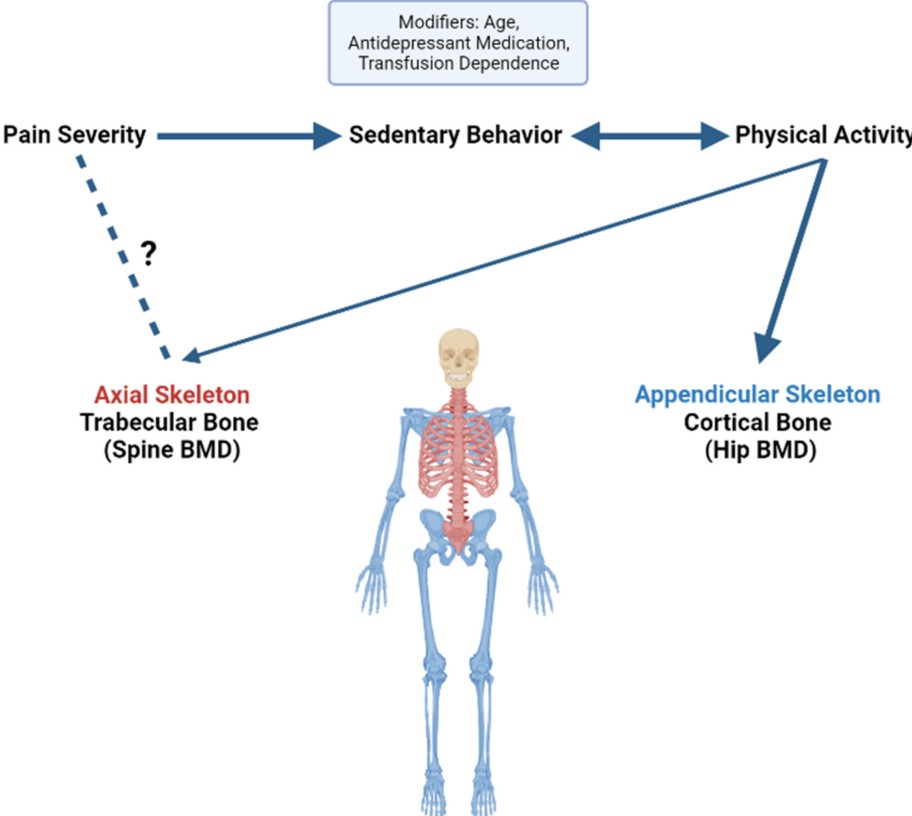

**Figure 3.** Hypothesized relationships among physical activity, pain, and bone health in patients with thalassemia. Pain severity was positively correlated with time spent in sedentary behavior. Sedentary behavior is typically inversely related to time spent participating in physical activity, and, in this cohort, increased physical activity was positively associated with bone mineral density (BMD) Z-score at the hip, a weight bearing skeletal site, while those who followed activity guidelines had higher spine BMD Z-scores. Pain severity was associated with spine BMD, potentially indicating sub-clinical vertebral compression fractures. Age, antidepressant usage and transfusion dependency were covariates which modulated the relationships among these variables.

Physical activity is difficult to measure objectively without the use of accelerometers. In this study, we chose to use two validated surveys to assess routine activity patterns. For the youth, the FNPA questionnaire was used to assess overall healthy behavior, including usual activity patterns. Although it is a validated tool, thresholds for total FNPA scores have not been established for determining healthy or unhealthy home environments. Therefore, interpretation of the data is somewhat subjective. Moreover, in the development of the FNPA, it was noticed that the total score was dependent upon the child and parent's BMI and ethnicity, as well as the family income. In this study, BMI was not related to FNPA score in Thal youth. Although parent BMI and family income were not collected, it is clear that the present cohort's mean FNPA score was significantly lower than that of the healthy controls for which the study was validated [24]. Given these findings, future interventions with Thal youth should include parents to ensure that healthy behaviors are encouraged in the home.

Only 37% of adults in the present cohort met the CDC guideline for physical activity, which is defined as a minimum of 30 min of moderate exercise 5 days per week or a total

of 150 min per week. Although sedentary behavior is also quite common in non-Thal US adults, given the high prevalence of low bone mass in thalassemia and the plausible effect of activity on bone health, exercise interventions are justified. There are only a handful of published exercise intervention studies in patients with Thal, and, given their short intervention period, none included bone health outcomes. In 2013, Arian and colleagues recruited 61 adult TDT patients and randomly assigned them to either an 8-week walking program or to engage in their standard level of activity [36]. Investigators observed an improvement in self-reported quality of life measures in the experimental cohort, which was attributed to exercise. More recently, 40 adult TDT patients were assigned to water exercise three times per week for 8 weeks [37]. Similar positive effects on quality of life were observed following this exercise regimen, reinforcing the potential beneficial effect of exercise in Thal.

This study had several limitations. Although validated questionnaires were used, there is always the possibility that self-reported physical activity and/or sedentary behaviors may not reflect actual activity patterns in these patients. The BPI-SF survey only focused on pain in the previous 24 h and may therefore not be reflective of chronic pain. The majority of the patients were TDT, which may reduce the generalizability of the results to NTDT patients. Fatigue-related anemia may be a significant determinant of sedentary behavior in NTDT; future studies could explore this in a larger cohort of NTDT patients. We also had no objective measures of iron toxicity or pretransfusion hemoglobin levels. Despite these limitations, robust relationships were observed among pain, activity, and bone health, which may be the basis for future interventions and research.

In conclusion, patients with Thal frequently experience pain and are prone to sedentary behaviors. These results suggest that decreased physical activity and increased sedentary behavior contribute to low bone mass in Thal. Interventional studies focused on increasing physical activity may not only reduce pain and improve quality of life, but may also have a significant positive effect on bone health, thereby reducing the long-term burden of osteoporosis.

**Author Contributions:** E.B.F. and E.K.G. conceptualized the study and submitted it for ethics review; R.M. recruited and obtained consent from the patients; E.B.F., E.K.G. and S.B. conducted the chart review, E.B.F. performed the analyses, E.B.F. and S.B. wrote the first manuscript draft; A.L. and E.K.G. assisted in the review and interpretation of the data. All authors have read and agreed to the published version of the manuscript.

**Funding:** This study was supported in part by an NIH training grant HL R25125451.

**Institutional Review Board Statement:** This study was approved by the Institutional Review Board at the UCSF Benioff Children's Hospital Oakland, protocol #2017-123.

**Informed Consent Statement:** Informed written consent was obtained from all subjects involved in the study, for subjects aged 5 to 17 years, assent was obtained.

**Conflicts of Interest:** The authors declare no conflict of interest.

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
