# Peer review of "Relationships among Physical Activity, Pain, and Bone Health in Youth and Adults with Thalassemia: An Observational Study"

_thalassrep, doi:10.3390/thalassrep12030014_

Round 1
Reviewer 1 Report
The authors describe the relationships between physical activity, pain, and bone Health in Youth and Adults with transfusion-dependent thalassemia (TDT) and non-transfusion-dependent thalassemia (NTDT) which is interesting for physicians who are working in this field. However, there are some issues that the authors should answer listed below
1- Methods: - Why the study was not designed case-control which the results would be more valuable?
-How is the sample size calculated?
- All the thalassemia patients do not seem to be TDT and it should be evaluated separately (TDT VS NTDT) since the factors that affected bone mass and osteoporosis are somehow different between these two groups.
- The age range is heterogenous from childhood to adult. How can the confounding factors separate youth and adult age or between TDT and NTDT? Moreover, the frequency and mechanism of osteoporosis in TDT are not the same as in NTDT. Therefore, I suggest separating these two types of thalassemia.
- Bone mineral density (BMD) Z-scores are used to detect a low bone mass. Is it validated in the pediatric age group? Why is T score not used?
2-Results: It should be divided the patients into TDT and NTDT and evaluated for physical activity, pain, bone mass, and bone Health separately.
For example, the authors mentioned that "transfusion dependent tended to have a higher prevalence of low bone mass compared to those who were not on chronic transfusion therapy (67% vs. 42%, p=0.1)" but it is not supported by the methods.
Minor comment: -Abstract: "71" should be written in letters at the beginning of the sentence
Author Response
Response to Reviewers Comments
Manuscript ID: thalassrep-1786752
Type of manuscript: Article
Title: Relationships between Physical Activity, Pain, and Bone Health in Youth and Adults with Thalassemia: an Observational Study
Authors: Ellen B Fung *, Elijah Goldberg, Sakina Bambot, Raquel Manzo,Ashutosh Lal
Reviewer #1
The authors describe the relationships between physical activity, pain, and bone health in Youth and Adults with transfusion-dependent thalassemia (TDT) and non-transfusion-dependent thalassemia (NTDT) which is interesting for physicians who are working in this field. However, there are some issues that the authors should answer listed below
Methods:
- Why the study was not designed case-control which the results would be more valuable?
Thank you for this comment. Indeed, a case-control design might have been a stronger study design, however, there was limited funding available for this project, therefore, an observational cohort study was chosen for this pilot project. Despite the limitations, we were encouraged by the observations and look forward to exploring some of the associations in future studies.
- How is the sample size calculated?
A sample size was not calculated for this pilot project as there were no previously collected data by which to base a sample size. Therefore, we conducted an observational cohort study with convenient sampling. The aim was to conduct the surveys in a minimum of 75 subjects, or roughly 50% of the total number of patients who routinely attend our thalassemia comprehensive clinic. We felt as though this size would give us a reasonable and representative sample by which to explore activity and pain patterns in our clinic sample.
- All the thalassemia patients do not seem to be TDT and it should be evaluated separately (TDT VS NTDT) since the factors that affected bone mass and osteoporosis are somehow different between these two groups.
Thank you for this observation. We chose to combine the TDT and NTDT patients in this observational analysis for 3 reasons, 1) from prior reports in a large sample of patients with Thal from North America we have found that low bone mass and osteoporosis affect both NTDT and TDT patients equally (Vogiatzi JBMR 2009;24:3:543-547), though we agree that the etiology of and main contributors to low bone mass may be different between the 2 groups, 3) we did not observe a difference in pain prevalence or severity, or vitamin D status by transfusion status and, 2) we had a small number of NTDT patients (n=13 or 18% of the total cohort) therefore we would have too few to draw any reasonable conclusions if we considered them as an independent group. However, knowing there are inherent differences between these 2 types of patients, we chose to include transfusion as a cofounding variable in our multivariate analysis. We understand that some readers may be interested in a few of the findings focused on the NTDT patients and therefore we have added a few lines to Table 1, and text to the results regarding the prevalence of low bone mass in TDT vs. NTDT by age group. We have also added a statement to our limitations, recognizing it would be valuable to include a larger cohort of NTDT patients in future studies.
- The age range is heterogenous from childhood to adult. How can the confounding factors separate youth and adult age or between TDT and NTDT? Moreover, the frequency and mechanism of osteoporosis in TDT are not the same as in NTDT. Therefore, I suggest separating these two types of thalassemia.
Answered above. We have included a break out of some of these variables in Table 1 for the reader.
- Bone mineral density (BMD) Z-scores are used to detect a low bone mass. Is it validated in the pediatric age group? Why is T score not used?
Thank you for this question. Yes, low bone mass is validated in the pediatric population and has been for nearly a decade. There are robust reference data to use in pediatrics starting at age 3 which are used to calculate Z-scores vs. T-scores. The International Society for Clinical Densitometry has published position statements regarding how to report information related to BMD in pediatrics (iscd.org). BMD Z-scores are to be presented, and not T-scores, up until the patient is 50 years of age. Low bone mass is defined as a Z-score < -2.0.
Results:
-It should be divided the patients into TDT and NTDT and evaluated for physical activity, pain, bone mass, and bone Health separately.
Answered this question above, additional information has been added to the text
-For example, the authors mentioned that "transfusion dependent tended to have a higher prevalence of low bone mass compared to those who were not on chronic transfusion therapy (67% vs. 42%, p=0.1)" but it is not supported by the methods.
We will attempt to address your concern though we are not exactly sure what is meant by ‘not supported by the methods.’ This is an observational study, as such presentation of prevalence and associations are valid statements. We refrained from the use of any reference to causality given this was not an interventional study design.
-Minor comment: -Abstract: "71" should be written in letters at the beginning of the sentence
Thank you for noticing this, we have made the change to the abstract
Reviewer 2 Report
The authors describes relationships between Physical Activity, Pain, and Bone Health in Youth and Adults with Thalassemia:
INTRODUCTIONS
The introduction is clear and well written
At line 79-80 the authors reported the sample of the patients comprise adult and pediatric patients while in the title of the manuscript are reported adults and youth patients plese specify
Materials and Methods and Statistics
These section is clear and well written
Results
At line 159-160 "Interestingly, vitamin D insufficiency was more common
160 in youth subjects compared to adult subjects (p=0.001)" How the authors could better explain these finding?
The sections "Result and discussion" are too long.
Author Response
Reviewer #2
The authors describe relationships between Physical Activity, Pain, and Bone Health in Youth and Adults with Thalassemia:
INTRODUCTIONS
-The introduction is clear and well written
Thank you for your comments.
-At line 79-80 the authors reported the sample of the patients comprise adult and pediatric patients while in the title of the manuscript are reported adults and youth patients please specify
Thank you for this oversight. Our pediatric cohort of patients include both children and adolescent subjects. Given the term youth encompasses both children and adolescents, we chose to go with that term. We have revised the sentence on line 79 to ‘Youth” instead of “Pediatric”
Materials and Methods and Statistics
-These section is clear and well written
Thank you for your comment.
Results
-At line 159-160 "Interestingly, vitamin D insufficiency was more common in youth subjects compared to adult subjects (p=0.001)" How the authors could better explain these finding?
Excellent question, and one we struggle with but have made note of to be mindful when we follow our adolescent patients in clinic. We did not observe a difference in vitamin D insufficiency by transfusion status (now included in Table 1), so this difference by age group could be explained by the adolescent subjects being less adherent to daily vitamin D supplementation. Though we do not have data to support this statement directly, we have found that adolescents have been less adherent to interventions when compared to adults with Thal in past studies (Fung, AJH 2012).
-The sections "Result and discussion" are too long.
Thank you for this comment. The other reviewer asked us to add more detail on the NTDT which we have done, though have shortened the original methods and discussion in response to your request.
Round 2
Reviewer 1 Report
The revised manuscript looks good and all questions have been addressed